# CDK control pathways integrate cell size and ploidy information to control cell division

James Oliver Patterson[1,2]*, Souradeep Basu[1]*, Paul Rees[2,3], Paul Nurse[1,4]

[1]Cell Cycle Laboratory, The Francis Crick Institute, London, United Kingdom; [2]College of Engineering, Swansea University, Swansea, United Kingdom; [3]Imaging Platform, Broad Institute of Harvard and MIT, Cambridge, United States; [4]Laboratory of Yeast Genetics and Cell Biology, Rockefeller University, New York, United States

**Abstract** Maintenance of cell size homeostasis is a property that is conserved throughout eukaryotes. Cell size homeostasis is brought about by the co-ordination of cell division with cell growth and requires restriction of smaller cells from undergoing mitosis and cell division, whilst allowing larger cells to do so. Cyclin-CDK is the fundamental driver of mitosis and therefore ultimately ensures size homeostasis. Here we dissect determinants of CDK activity in vivo to investigate how cell size information is processed by the cell cycle network in fission yeast. We develop a high-throughput single-cell assay system of CDK activity in vivo and show that inhibitory tyrosine phosphorylation of CDK encodes cell size information, with the phosphatase PP2A aiding to set a size threshold for division. CDK inhibitory phosphorylation works synergistically with PP2A to prevent mitosis in smaller cells. Finally, we find that diploid cells of equivalent size to haploid cells exhibit lower CDK activity in response to equal cyclin-CDK enzyme concentrations, suggesting that CDK activity is reduced by increased DNA levels. Therefore, scaling of cyclin-CDK levels with cell size, CDK inhibitory phosphorylation, PP2A, and DNA-dependent inhibition of CDK activity, all inform the cell cycle network of cell size, thus contributing to cell size homeostasis.

*For correspondence:
jamesop@gmail.com (JOP);
saz.basu@crick.ac.uk (SB)

**Competing interests:** The authors declare that no competing interests exist.

## Introduction

Cells display homeostatic behaviour in maintaining population cell size by controlling cell size at mitosis (*Fantes et al., 1975*; *Ginzberg et al., 2015*; *Wood and Nurse, 2015*; *Lloyd, 2013*). This homeostasis is driven by larger cells being more likely to divide than smaller cells, resulting in the correction at cell division of cell size deviances (*Fantes et al., 1975*; *Fantes, 1977*; *Patterson et al., 2019*). Cyclin-dependent kinase (CDK^Cdc2) is the master regulator of mitosis and cell division, and therefore the propensity for smaller cells not to divide must ultimately feed into the regulation of CDK activity (*Coudreuse and Nurse, 2010*).

CDK activity is subject to several mechanisms of control: cyclin synthesis and subsequent binding of cyclin to CDK, which drives CDK into a catalytically competent form *Solomon et al., 1990*; Wee1 kinase and Cdc25 phosphatase act to inhibit or activate CDK, respectively, through regulatory tyrosine phosphorylation (*Nurse, 1975*; *Russell and Nurse, 1986*; *Gould and Nurse, 1989*); and PP2A phosphatase works to remove phosphates deposited by CDK reducing its net activity (*Kinoshita et al., 1990*; *Kinoshita et al., 1993*; *Gharbi-Ayachi et al., 2010*; *Mochida et al., 2009*; *Mochida et al., 2010*; *Mochida et al., 2016*), and also controls the phosphorylation state of Wee1 and Cdc25 to regulate the level of CDK tyrosine phosphorylation (*Lucena et al., 2017*; *Hutter et al., 2017*; *Rata et al., 2018*; *Kamenz et al., 2021*). Finally, the CDK control network also

co-ordinates cell division with cell growth through an unknown mechanism that responds to cell ploidy, with cell size generally doubling as ploidy doubles (*Wood and Nurse, 2015*).

It is likely that potential size control pathways will be integrated at the level of CDK activity control because CDK activity is the driver of mitosis. For example, in the fission yeast *Schizosaccharomyces pombe*, it has been proposed that size control was mediated by the DYRK kinase Pom1, which ultimately inhibits mitotic onset by causing the inhibitory tyrosine phosphorylation of CDK by signalling through the Wee1 kinase (*Martin and Berthelot-Grosjean, 2009*; *Moseley et al., 2009*). However, in both the absence of Pom1 itself or the absence of inhibitory tyrosine phosphorylation, cells are able to maintain cell size homeostasis (*Coudreuse and Nurse, 2010*; *Wood and Nurse, 2013*). Thus, there must exist alternative mechanisms by which fission yeast cells integrate cell size information into the CDK control network.

Much of our understanding of cyclin-CDK network regulation has been derived from in vitro studies, but these have limitations when considering cellular parameters such as cell size (*Mochida et al., 2016*; *Pomerening et al., 2005*; *Pomerening et al., 2003*; *Sha et al., 2003*). Here, therefore, we have studied in vivo regulation of cyclin-CDK activation at mitosis in the fission yeast. Using a novel CDK activity sensor, we have monitored cell size, CDK activity, and cyclin-CDK complex level simultaneously, whilst genetically varying regulators of the cyclin-CDK control system. We propose that CDK activity regulation through inhibitory tyrosine phosphorylation and PP2A work synergistically to communicate information about cell size to the CDK control network. Furthermore, we show that cyclin-CDK complex level scales with cell size, and this aids in the prevention of division in small cells. Finally, we show that in cells lacking PP2A and inhibitory tyrosine phosphorylation, haploid and diploid cells of equivalent size and similar cyclin-CDK concentration have differing cyclin-CDK activities, with diploid cells exhibiting a lower activity. This suggests that cyclin-CDK activity is increased in cells of lower ploidy. These experiments inform our understanding of the regulation of cyclin-CDK and illuminate how cell size is integrated into this regulatory network.

## Results

Given the complexity of the CDK regulatory network, we have used fission yeast cells containing a reduced CDK control system, with the cell cycle driven by a monomeric cyclin-CDK fusion-protein (C-CDK) (*Coudreuse and Nurse, 2010*). This simplifies the CDK control network by eliminating cyclin binding to CDK as a regulatory component and allows co-expression of both cyclin and CDK from a single promoter. This C-CDK fusion-protein is expressed under the Cyclin B$^{Cdc13}$ promoter, and therefore C-CDK expression mimics endogenous Cyclin B expression. Using this system, inhibitory Wee1-dependent phosphoregulation can also be removed using a non-phosphorylatable C-CDK$^{AF}$ mutant (*Coudreuse and Nurse, 2010*; *Wood and Nurse, 2013*). These C-CDK$^{AF}$ strains are healthy and viable, but have markedly distinct cell cycle profiles from C-CDK$^{WT}$ expressing strains, as they spend a significantly longer period in G1 than C-CDK$^{WT}$ cells (*Coudreuse and Nurse, 2010*). Nevertheless, C-CDK$^{AF}$ cells co-ordinate cell division with cell growth and maintain cell size homeostasis (*Figure 1a*; *Wood and Nurse, 2013*).

To examine the relationship between cell size, C-CDK concentration, and mitosis, we performed quantitative fluorescence time-lapse microscopy on strains expressing C-CDK$^{WT}$ and C-CDK$^{AF}$ fluorescently tagged with YFP (*Figure 1a–e*, *Figure 1—figure supplement 1*, *Figure 1—figure supplement 2a*). This analysis showed clear oscillations of C-CDK$^{WT}$ and C-CDK$^{AF}$, with degradation of C-CDK occurring just before cell division (*Figure 1b*). C-CDK$^{AF}$ oscillations were more variable, and 5% of the C-CDK$^{AF}$ cells trigger C-CDK degradation in the absence of division (*Figure 1—figure supplement 2*), similar to what has been observed in CDK1$^{AF}$ expressing human cells (*Pomerening et al., 2008*). In both backgrounds, C-CDK concentration scaled with cell size, with C-CDK$^{WT}$ exhibiting a higher amount of C-CDK at mitotic entry compared to C-CDK$^{AF}$ (*Figure 1c*). On investigating the links between the probability of a given cell to divide, cell size, and C-CDK level, we found that for C-CDK$^{WT}$ both cell size and C-CDK level reach sharp thresholds at which cell division rates increase (*Figure 1d,e*). In the absence of tyrosine phosphorylation, a sharp threshold for C-CDK$^{AF}$ levels still is present (*Figure 1e*), but occurs at a lower level than C-CDK$^{WT}$. C-CDK$^{AF}$ cells fail to generate a sharp threshold for cell size, but even without a clear size threshold, C-CDK$^{AF}$ cells still restrict smaller cells from division (*Figure 1d*).

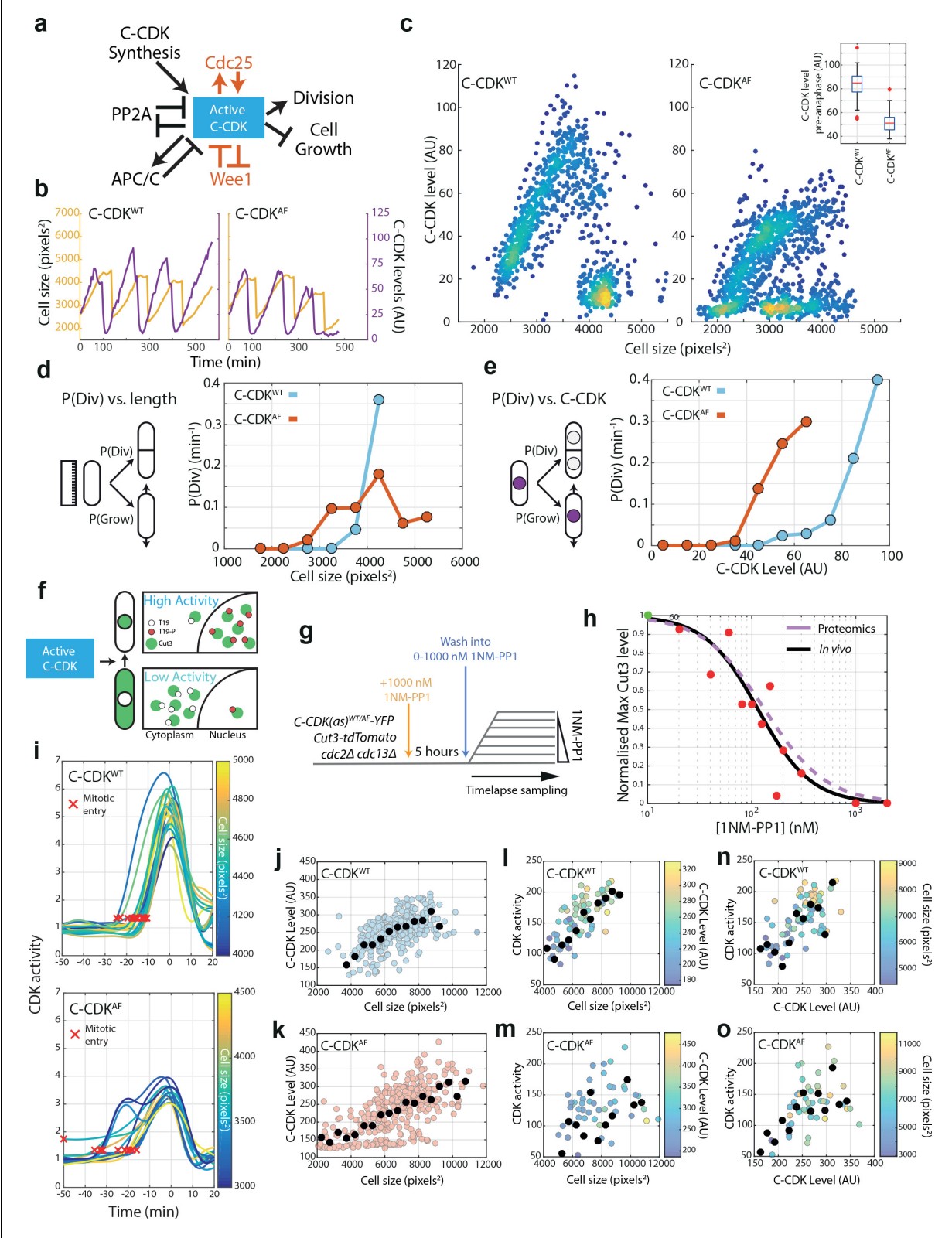

**Figure 1.** Cell size and C-CDK concentration dictate probability of division and CDK activity in C-CDK^WT and C-CDK^AF cells. (a) Schematic of major components influencing C-CDK activity at mitosis, and in red, the pathways that do not influence C-CDK^AF. The negative relationship between C-CDK activity and cell growth refers to the block of cell length extension in mitosis. PP2A opposes CDK activity by dephosphorylating CDK substrates, and also by opposing the activation of CDK at mitosis by opposing the phosphorylation of Wee1 and Cdc25. Reciprocally, CDK causes the downregulation

*Figure 1 continued on next page*

*Figure 1 continued*

of PP2A activity in mitosis. (**b**) Example cell lineage traces from time-lapse microscopy. Cell size in pixels$^2$ is given in orange, and C-CDK-YFP fluorescence intensity is given in purple. Steep decreases in cell size traces correspond to cell division. (**c**) Scatter plot of mean C-CDK level vs cell size from time-lapse microscopy data. C-CDK level is a measure of C-CDK-YFP fluorescence intensity. Colours indicate density of data. Inset boxplot is mean nuclear C-CDK concentration immediately prior to degradation at anaphase. Boxes represent interquartile range, with whiskers delimiting 5th–95th percentiles. C-CDK$^{WT}$ n = 28, C-CDK$^{AF}$ n = 44 full cycles. (**d**) Plot of the probability of division at the next timepoint (P(Div)) vs cell length for CDK$^{WT}$ and CDK$^{AF}$. Cells were followed through time-lapse microscopy with measurements taken each frame. P(Div) defined as the proportion of cells that undergo C-CDK degradation at anaphase by the next timepoint, given as rate per minute. Points represent cells binned by size, with points plotted at bin centre. C-CDK$^{WT}$ n = 685, C-CDK$^{AF}$ n = 961 timepoints. (**e**) Plot of P(Div) function vs C-CDK level for CDK$^{WT}$ and CDK$^{AF}$. C-CDK$^{WT}$ n = 685, C-CDK$^{AF}$ n = 961 timepoints. C-CDK-YFP intensity measurements taken every frame from time-lapse microscopy, and binned by C-CDK level. (**f**) Schematic of Cut3 as a CDK activity reporter. Mitotic CDK-dependent phosphorylation of Cut3 on T19 results in nuclear translocation of the protein. (**g**) Experimental outline of block and release time-lapse experiment for panels (**h, j–o**). Asynchronous cells possessing an analogue sensitive (as) CDK were blocked in G2 using 1 μM 1NM-PP1 for 5 hr and then released into a range of 1NM-PP1 concentrations. Cells were then followed and monitored for their Cut3-tdTomato nuclear/cytoplasmic (N/C) ratio (C-CDK activity) and C-CDK-YFP level using fluorescence time-lapse microscopy (see Materials and methods). Data for (**l–o**) were acquired 15 min following release from 1NM-PP1. (**h**) Maximum CDK activity (normalized against maximum level, obtained by release into DMSO) against 1NM-PP1 concentration. Red points are the median of the data sets for each drug concentration (N = 324), and green point is median in DMSO. Black line is the Hill equation fit to the median data by a nonl-inear fitting algorithm (IC50 = 115.4, Hill coefficient = −1.71). Purple dashed line is Hill curve derived from *Swaffer et al., 2016* dose–response data (IC50 = 133.4, Hill coefficient = −1.47). (**i**) Time-lapse quantification of CDK activity in asynchronous cells. Traces are aligned so that 0 min corresponds to peak Cut3-tdTomato N/C ratio. Curve smoothing could move Cut3 peak earlier/later than exactly 0 min. Trace colour indicates cell size. Red X indicates automatically defined mitotic entry point. C-CDK$^{WT}$n = 23 and C-CDK$^{AF}$n = 14. (**j**) Scatter plot of C-CDK-YFP levels against cell size. Experiment described in (**g**), with measurements taken before release from 1NM-PP1 block. Black points indicate binned data, bin window size 500 pixels$^2$. n = 324. Pearson correlation coefficient: 0.55. (**k**) As in (**j**), but with C-CDK$^{AF}$, n = 312. Pearson correlation coefficient: 0.62. (**l**) Scatter plot of peak Cut3-tdTomato level vs cell size. Experiment described in (**g**), with measurements taken 20 min after release from 1NM-PP1 block into DMSO. Black points indicate binned data, bin window size 500 pixels$^2$. Points are coloured by YFP C-CDK levels at release. n = 83. $R^2$ = 0.5040. Pearson correlation coefficient: 0.50. (**m**) As in (**l**), but with C-CDK$^{AF}$, n = 81. $R^2$ = 0.2150. Pearson correlation coefficient: 0.22. (**n**) Scatter plot of peak Cut3-tdTomato level vs C-CDK-YFP intensity level 20 min after release from 1NM-PP1 block into DMSO. Black points indicate binned data, bin window size 15 AU. Points are coloured by cell size at release. n = 83. $R^2$ = 0.3668. Pearson correlation coefficient: 0.60. (**o**) As in (**n**), but with C-CDK$^{AF}$, n = 81. $R^2$ = 0.5501. Pearson correlation coefficient: 0.74.

The online version of this article includes the following figure supplement(s) for figure 1:

**Figure supplement 1.** Automated image analysis pipeline for wide-field imaging.

**Figure supplement 2.** Fluorescence time-lapse quantification of C-CDK dynamics in unperturbed cell cycles.

**Figure supplement 3.** A time-lapse block and release assay to measure the effect of CDK inhibition on CDK activity in single cells.

**Figure supplement 4.** Cut3-GFP as a marker of CDK activity in WT and AF cell strains.

**Figure supplement 5.** An imaging flow cytometry assay reveals that size, C-CDK level, and tyrosine phosphorylation dictate the rate and timing of CDK activation at mitosis.

**Figure supplement 6.** Size-dependent grading of mitotic entry rates and timing are dose responsively dependent on CDK inhibition.

C-CDK level is not a direct measure of C-CDK activity because of the multiple regulatory networks affecting CDK (*Pomerening et al., 2005*). To investigate CDK activity, cell size, and C-CDK level at the same time, we developed an in vivo single-cell assay of CDK activity. We used Cut3, the Smc4 homolog, as a CDK activity biosensor, because it translocates from the cytoplasm into the nucleus upon CDK-dependent phosphorylation of a single site in its N-terminus (*Figure 1f*; *Sutani et al., 1999*). Thus, the Cut3 nuclear/cytoplasmic (N/C) ratio can be used to assess CDK activity, a method that has been applied to other protein kinases (*Spencer et al., 2013*; *Araujo et al., 2016*). As a test of this assay, we blocked cells expressing fluorescently tagged Cut3 in the background of a bulky ATP-analogue-sensitive C-CDK (*Bishop et al., 2000*) using 1NM-PP1, and tracked single cells following their release from G2 arrest into a range of 1NM-PP1 doses (*Figure 1g*, *Figure 1—figure supplement 3*). The response of the maximum nuclear Cut3 concentration to 1NM-PP1 was similar to the one measured in our previous phosphoproteomics study (*Swaffer et al., 2016*), confirming that the sensor reflects in vivo CDK activity (*Figure 1h*). Subsequently, we examined CDK activity in unperturbed cells measured by the Cut3 N/C ratio and showed that it both rises to a higher level in C-CDK$^{WT}$ cells in comparison to C-CDK$^{AF}$ cells and also that progress through mitosis in C-CDK$^{AF}$ cells is slower and more variable (*Figure 1i*, *Figure 1—figure supplement 4*).

We next investigated the links between C-CDK protein levels, CDK activity, and cell size in C-CDK$^{WT}$ and C-CDK$^{AF}$ cells, which have been enlarged beyond their physiological cell size. During a G2/M block (*Figure 1g*), cell size and C-CDK enzyme concentration (as measured by C-CDK-YFP

fluorescence intensity) scaled with each other in both backgrounds (*Figure 1j,k*). After the release from CDK inhibition, C-CDK$^{WT}$ activity correlated well with both cell size and C-CDK protein level (*Figure 1l,n*). However, peak C-CDK$^{AF}$ activity correlated better with protein level than with cell size (*Figure 1m,o*). The link between cell size and CDK activity was much clearer for C-CDK$^{WT}$ than for C-CDK$^{AF}$ in these low-throughput time-lapse assays (*Figure 1m*). Therefore, we repeated this experiment using a high-throughput assay based on imaging flow cytometry (*Figure 1—figure supplements 5* and *6*) and observed that peak CDK activity in both C-CDK$^{AF}$ and C-CDK$^{WT}$ was clearly size dependent (*Figure 1—figure supplement 5e*). Thus, CDK tyrosine phosphorylation helps to inform the cell division machinery of cell size (*Figure 1d,l*). However, in the absence of tyrosine phosphorylation, C-CDK$^{AF}$ cells are still able to generate a threshold C-CDK level for division and prevent small cells from division (*Figure 1e,o*, *Figure 1—figure supplement 5e*).

A complication of the above assay is that cell size scales with C-CDK level (*Patterson et al., 2019*; *Coudreuse and Nurse, 2010*; *Navarro and Nurse, 2012*; *Figure 1c,j,k*). To uncouple cell size from C-CDK level, and study if small cells are prevented from entering mitosis due to low C-CDK level or for some other reason, we developed a more flexible single-cell CDK assay system. This assay was also based on Cut3 translocation into the nucleus (*Figure 2a*) but used a brighter synthetic C-CDK activity sensor, synCut3-mCherry to allow its co-detection with C-CDK in a high-throughput assay (*Figure 2—figure supplement 1*). This sensor was expressed in a strain where the endogenous CDK network can be switched off using a temperature-sensitive CDK1 allele, *cdc2$^{TS}$*. A tetracycline-inducible C-CDK tagged with superfolder GFP (sfGFP) was constructed and made non-degradable (*Yamano et al., 1998*) as well as sensitive to inhibition by 1NM-PP1. Induction of C-CDK at the *cdc2$^{TS}$* restrictive temperature allows the study of the activity of the inducible C-CDK without either wild-type CDK activity or C-CDK-sfGFP proteolysis during mitosis. Using this assay, we acquired hundreds of thousands of images of single cells, which allowed us to study the in vivo biochemistry of CDK activity in response to a wide range of C-CDK concentrations in physiologically sized cells. C-CDK level was uncoupled from cell size as induction of C-CDK was not dependent on cell size (*Figure 2b,c*). Results from this assay demonstrated that in vivo CDK activity was dependent on C-CDK level and was reduced when CDK activity was inhibited using 1NM-PP1 (*Figure 2d*, *Figure 2—figure supplement 2*).

Combining this system with genetic backgrounds in which major C-CDK regulation was altered, we analysed how mechanisms of CDK regulation affected C-CDK activity in relation to cell size. We performed the assay in backgrounds lacking the major PP2A catalytic subunit (PP2A$^{ppa2}$Δ) (*Kinoshita et al., 1990*; *Kinoshita et al., 1993*), inhibitory CDK tyrosine phosphorylation, or both (*Figure 2e*). Following endogenous CDK1 inactivation after temperature shift, both PP2A$^{+}$ and PP2AΔ cells arrest in an almost uniform G2 state, ensuring that downstream analysis is not confounded by cells arresting in different phases of the cell cycle (*Figure 2f*). C-CDK levels increased similarly upon induction in all mutant backgrounds (*Figure 2g*). Population mean C-CDK activity was comparable between all conditions (*Figure 2h*); however, C-CDK activity displayed differences at the single-cell level when activity was measured in cells of different sizes. In all genetic backgrounds at the same level of C-CDK enzyme, maximum C-CDK activity increases with cell size (*Figure 2i*). This is particularly noticeable when directly comparing the maximum C-CDK activity of cells with a C-CDK level of ~750 AU in the 8 µm bin to the 14 µm bin in all backgrounds (*Figure 2i*, dashed lines). The single-cell dose–response of CDK activity on C-CDK$^{WT}$ concentration background is clearly bistable, with cells existing in either an 'on' or an 'off' state. The mean CDK activity is relevant directly for strains expressing C-CDK$^{AF}$, as these cells exhibit little bistability in CDK activation. In the C-CDK$^{WT}$ expressing cells, there are two population distributions demonstrating bistability. We averaged the two population means as the gradient of this line shows the degree of bifurcation between the lower and the upper CDK activity populations. The C-CDK concentration required to switch cells 'on' decreases with increasing cell size, and the sharpness of the transition increases with size (*Figure 2i,k*). This bistable behaviour is heavily dependent on CDK tyrosine phosphorylation (*Figure 2i,k,l*). Removal of PP2A allows the attainment of the 'on' state at lower cell sizes (*Figure 2i*), effectively shifting the C-CDK dose–response curve towards lower sizes without altering the shape of the response (*Figure 2k*). In addition, PP2A also adds switch like behaviour to the C-CDK activity dose-response, as bistable behaviour with C-CDK$^{AF}$ is not present with C-CDK$^{AF}$ PP2AΔ (*Figure 2i* dashed box, inset and *Figure 2l*).

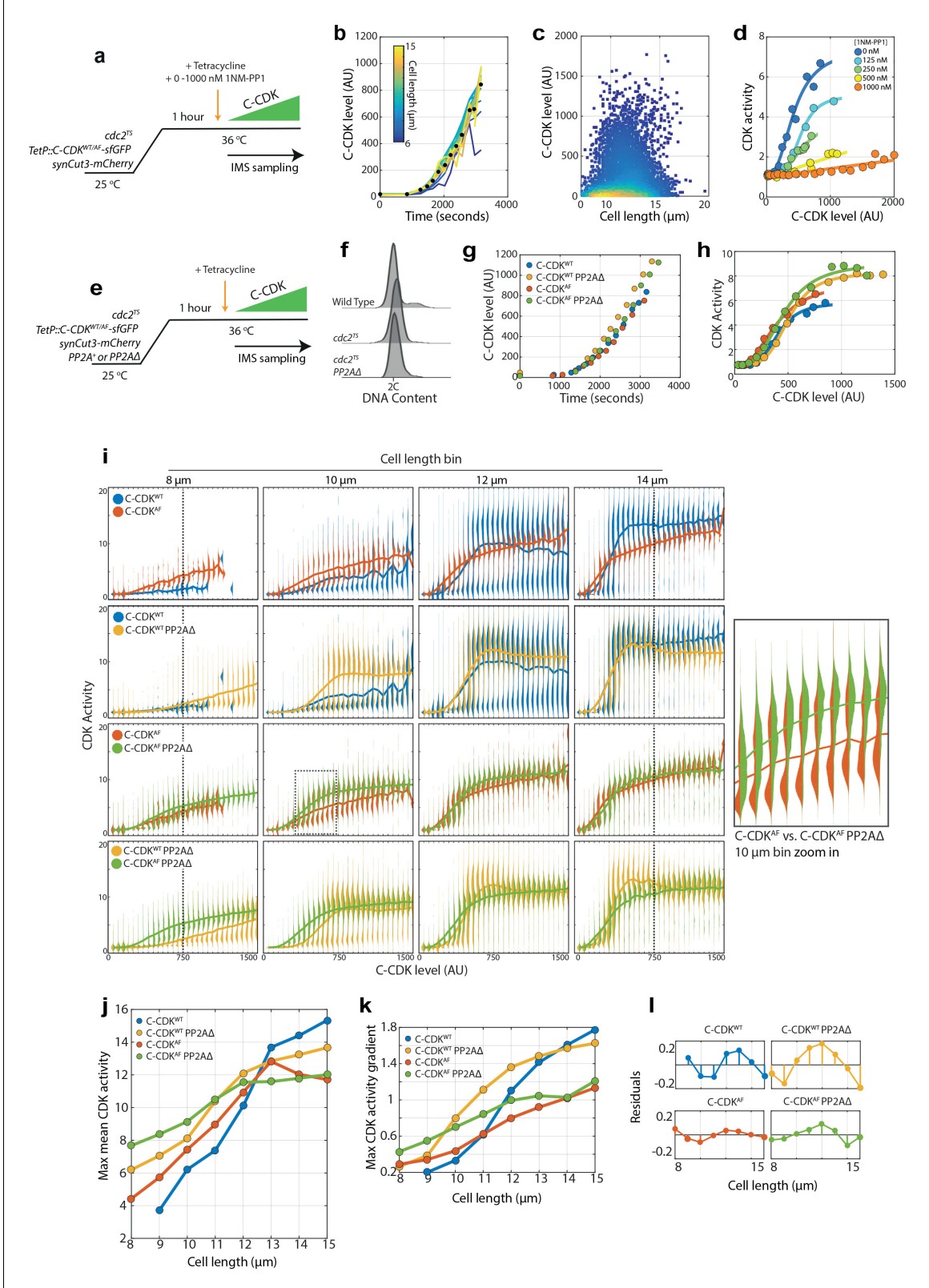

**Figure 2.** Cell size is able to modulate CDK activity independently of canonical CDK regulation. (**a**) Experimental outline for figure for (**b–d**). Cells were held at 36°C for 1 hr to ablate the function of the temperature-sensitive (TS) *cdc2* allele. C-CDK-sfGFP expression was induced by addition of tetracycline, and ectopic C-CDK concentration and CDK activity were measured by sequential sampling during induction. Induced C-CDK-sfGFP lacks its degron box sequence, and therefore is not degraded at anaphase. Sequential sampling during C-CDK-sfGFP induction begins at the point of

*Figure 2 continued on next page*

Figure 2 continued

tetracycline addition, with roughly one sample taken every 3 min after the start of C-CDK production. Sampling is conducted using an imaging flow cytometer (IMS). (**b**) Expression of C-CDK^WT^ from point of tetracycline addition. Different coloured lines represent different size bins. Black dots represent mean C-CDK-sfGFP level over all size bins for given timepoint. After lag period of ~1000 s after tetracycline addition, samples are taken roughly every 3 min. n = 759,633. (**c**) Scatter plot of cell length vs C-CDK-sfGFP levels. Coloured by density of data points. Data collected throughout induction. n = 759,633. (**d**) Mean CDK activity dose–response against C-CDK-sfGFP in the presence of annotated levels 1NM-PP1. Circles represent average CDK activities across all cells from a single sample taken after induction. 0 nM n = 166,081; 125 nM n = 60,759; 250 nM n = 165,128; 500 nM n = 135,670; and 1000 nM n = 231,995. (**e**) Experimental outline for (**f**–**k**). Cells were held at 36°C for 1 hr to ablate *cdc2^TS^* function. After 1 hr, C-CDK^WT^ or C-CDK^AF^ fused to sfGFP was induced with tetracycline in cells with either the major PP2A catalytic subunit (encoded by the *ppa2* gene) deleted or present. Induced C-CDK-sfGFP lacks its degron box sequence, and therefore is not degraded at anaphase. Sequential sampling during C-CDK-sfGFP induction begins at the point of tetracycline addition, with timepoints taken roughly every 3 min after 1000s lag period in C-CDK-sfGFP induction. (**f**) Flow cytometric DNA content analysis for wild-type cells, *cdc2-M26* cells, and *cdc2-M26 PP2AΔ* cells. The major PP2A, *ppa2*, was deleted in PP2AΔ cells. Cells were fixed for sampling after the block lengths specified in (**e**), before the addition of tetracycline. (**g**) Induction of C-CDK after tetracycline addition. Points represent mean concentration of C-CDK-sfGFP across all size bins at indicated timepoints. CDK^WT^n = 166,081. C-CDK^WT^ PP2AΔ n = 175,247. C-CDK^AF^n = 177,292. C-CDK^AF^ PP2AΔ n = 174,847. (**h**) C-CDK activity against C-CDK-sfGFP level in given genetic backgrounds defined in (**g**). Points represent mean C-CDK activity of all cells. Data is pooled from experiment in (**e**), from all timepoints following tetracycline induction. Key is the same as (**g**). (**i**) Violin plots of single-cell C-CDK-sfGFP level against CDK activity in annotated size bins and strain backgrounds. Solid line through violin plot indicates the mean CDK activity within the C-CDK level bin. (**j**) Maximum mean CDK activity vs cell length in annotated strain backgrounds. Max mean CDK activity is the maximum mean CDK activity within a C-CDK fluorescence level bin for a given cell size. The mean CDK activity level across all fluorescence bins is shown by the solid line in the violin plots in (**i**). (**k**) Maximum gradient of the mean lines in (**i**) plotted against cell length. Maximum gradient of change is derived from a spline fit to the mean CDK activity vs C-CDK-sfGFP level trace. (**l**) Linear regression lines were fit to data in (**k**), and residuals were plotted (actual value – predicted value). Non-linear residuals indicate bistability in CDK activation.

The online version of this article includes the following figure supplement(s) for figure 2:

**Figure supplement 1.** A new synthetic CDK sensor for *S. pombe*.
**Figure supplement 2.** A single-cell in vivo biochemistry approach permits decoupling of cell size from C-CDK concentration.

When looking across all size bins, maximum C-CDK activity increases with cell size in all genetic backgrounds, but plateaus at about 12–13 µm in the absence of tyrosine phosphorylation (*Figure 2j*). However, it is clear that cell size is able to regulate C-CDK activity even in the absence of both tyrosine phosphorylation and PP2A (*Figure 2i,j*). These results are consistent with our previous observations (*Figure 1*), that although tyrosine phosphorylation has a role in informing the cell cycle machinery of cell size, small cells are still restricted from mitosis when tyrosine phosphorylation is absent.

Inhibitory tyrosine phosphorylation directly results in a reduction of intrinsic CDK activity (*Berry and Gould, 1996*), whilst PP2A has a dual mechanism of CDK activity modulation: PP2A is able to regulate inhibitory tyrosine phosphorylation by controlling the phosphorylation state of Wee1 and Cdc25 (*Lucena et al., 2017*), and in addition can directly oppose the phosphorylation of CDK substrates (*Godfrey et al., 2017*). We therefore sought to calculate the contributions of tyrosine phosphorylation and PP2A in restricting CDK activity, both in contexts with and without tyrosine phosphorylation. This was carried out to examine if their combined contribution was greater than the sum of their parts. To calculate the individual contributions of tyrosine phosphorylation and PP2A in restricting C-CDK activity, first we measured the threshold C-CDK level required for 50% of cells to reach a C-CDK activity determined as being >5 in arbitrary units (see *Figure 3* legend) in different strain backgrounds within different size bins (*Figure 3a*). This value was chosen as an approximate value of the C-CDK concentration required in vivo to trigger mitotic entry in wild-type cells (*Figure 1i*). When this C-CDK threshold level was plotted across all size bins (*Figure 3b*), the threshold was seen to be size dependent in all strain backgrounds, with wild-type cells exhibiting the strongest capacity to raise the C-CDK level threshold for mitosis in smaller cells. By subtracting the curves of cell length vs mitotic C-CDK level (*Figure 3c*) for various backgrounds, we were able to estimate the contributions of tyrosine phosphorylation and PP2A in a given background. For example, C-CDK^WT^ PP2AΔ–C-CDK^AF^ PP2AΔ estimates the ability of tyrosine phosphorylation alone to restrict mitotic entry in a background lacking PP2A. Inhibitory tyrosine phosphorylation is able to restrict cells with 600 units of C-CDK from entering mitosis at 8 µm cell length, but only 200 units of C-CDK at 10 µm (*Figure 3c*, yellow). If the different components of the CDK control network act separately, adding individual threshold contributions together would generate a threshold curve similar to the wild-type curve. However, when the contributions of tyrosine phosphorylation and PP2A were added

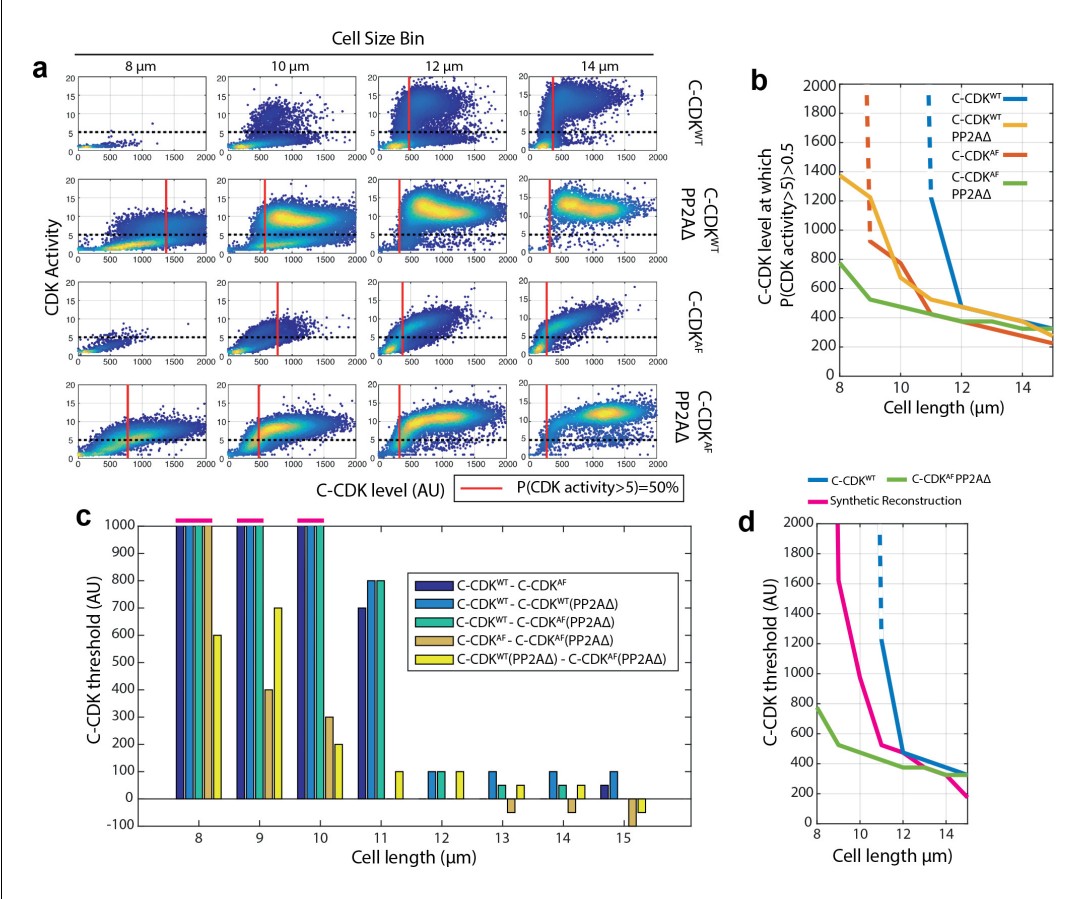

**Figure 3.** CDK tyrosine phosphorylation and PP2A act synergistically to restrict division in small cells. (**a**) Scatter plots of C-CDK level against CDK activity. Either C-CDK^WT or C-CDK^AF fused to sfGFP was induced in backgrounds with PP2A either lacking or present. *PP2AΔ* refers to a deletion of the *ppa2* gene. Red line indicates the C-CDK-sfGFP level at which 50% of cells have a CDK activity greater than 5. Black dashed line marks CDK activity of 5. Data taken from *Figure 2i*. (**b**) C-CDK-sfGFP level at which 50% of cells have C-CDK activity of >5. Data is taken from (**a**) across all size bins. Y-axis represents the C-CDK-sfGFP threshold at which 50% of cells will have a C-CDK activity of 5. Dashed lines indicate values where this C-CDK-sfGFP threshold level is undefined due to the threshold being unattainable in experimental conditions. (**c**) Piecewise dissection of the amount of C-CDK-sfGFP a particular component of the cell cycle network is able to prevent from switching to an 'on' state (C-CDK activity level of 5) in different size bins. Bar chart shown is of subtractions of curves described in key (inset). For example, C-CDK^WT–C-CDK^AF gives the C-CDK threshold tyrosine phosphorylation alone (in a background with PP2A present) is able to generate to restrict C-CDK activation. Values that are undefined due to undefined original threshold values from (**a**) are taken to be 1000 units and are marked above the axis (pink). (**d**) Cell length against C-CDK level threshold of annotated curves. Here, a synthetic threshold curve is built (pink), by adding the individual component regulatory contributions of CDK tyrosine phosphorylation (**c**, yellow) and PP2A (**c**, orange) to the base curve of C-CDK^AF PP2AΔ (green) to try and re-capitulate the WT behaviour (blue). Dashed line indicates undefined threshold values.

to the C-CDK^AF PP2AΔ curve, they did not re-capitulate the wild-type curve (*Figure 3d*). Thus, this analysis suggests that there is synergy between the tyrosine phosphorylation network and PP2A activity and that this synergy is important for establishing the C-CDK level threshold for division.

We have shown that small cells are normally prevented from division by their low C-CDK protein level (*Figure 1*), along with PP2A and tyrosine phosphorylation working synergistically to increase the level of C-CDK needed to trigger division in smaller cells (*Figure 3*). Strikingly however, in the absence of these major regulators, small cells are still able to restrict division by lowering CDK activity as a result of some other factor related to cell size (*Figure 2h,i,j*). This unknown factor is able to lower CDK activity in small cells despite high C-CDK levels, thus restricting them from division (*Figure 2i*).

Given the positive relationship between maximum C-CDK activity and increasing cell size in the C-CDK^AF PP2AΔ mutant (*Figure 2i*), we hypothesized that cells dilute a CDK inhibitor as they grow (*Fantes et al., 1975*), perhaps through a titration-based mechanism. Cell size is linked to ploidy

through an unknown mechanism, and so we tested whether DNA concentration could influence CDK activity, and therefore be a candidate for the unknown factor able to lower C-CDK activity in small cells. We induced C-CDK$^{AF}$ in haploid and diploid variants of the C-CDK$^{AF}$ PP2AΔ strain, thereby eliminating all major CDK regulation at mitosis (*Figure 4a*). Both haploid and diplod cells were present almost uniformly in G2 after endogenous CDK1 inactivation, as cells with 2C and 4C DNA content respectively (*Figure 4b*), and expressed C-CDK$^{AF}$-sfGFP in a largely size-independent manner (*Figure 4c*). Strikingly, diploid cells exhibited lower C-CDK activity in response to the same C-CDK enzyme concentration as haploids (*Figure 4d*). The EC50 of the diploid dose–response curve was almost double that of the haploid (*Figure 4e*). Looking at single-cell, volume-resolved data, the inhibition of C-CDK activity is most marked in smaller diploid cells, with larger diploid cells having almost indistinguishable dose–response curves from their haploid equivalents (*Figure 4f*). The effect of cell size on CDK activation is much less marked in larger than normal haploids (*Figure 4g*). The diploids, which feature cells of physiological diploid size, still experience DNA concentration-dependent inhibition of their CDK activity. The effect of equal C-CDK levels resulting in lower C-CDK activity in small diploids when compared to equivalent sized haploids is readily seen from the raw images (*Figure 4h*). Therefore, cells of different ploidies, but otherwise equivalent volume, experience variable CDK activity in response to equal C-CDK level. This suggests that even in the absence of all major CDK regulation, DNA concentration is able to lower CDK activity and prevents division in small cells. At higher volumes, this inhibition of CDK activity disappears, and so the regulation may operate through titrating out an inhibitor.

## Discussion

The cyclin-CDK complex, and its role in controlling mitotic onset, has been studied in many model eukaryotes from yeast (*Nurse et al., 1976*), through marine invertebrates (*Evans et al., 1983*), to mammalian cells (*Santos et al., 2012*). A number of regulatory components have been shown to be conserved across these model systems, including the CDK-activating cyclin B, inhibitory CDK tyrosine phosphorylation, and the CDK-counteracting PP2A phosphatase which both opposes CDK substrate phosphorylation and regulates CDK inhibitory phosphorylation through the Wee1/Cdc25 control loop. Despite extensive study, these studies have yet to reveal a fully satisfactory mechanism for cell size homeostasis at the onset of mitosis. To improve our understanding of this control system, we have focused on how CDK activity itself is directly regulated in the context of cell size. Our approach has demonstrated that three mechanisms inform the cell cycle control network of cell size through CDK activity control: C-CDK enzyme concentration scaling with cell size, synergistic PP2A and tyrosine-phosphorylation-dependent C-CDK threshold scaling, and DNA concentration-dependent inhibition of CDK activity. Our results demonstrate that C-CDK activity vs C-CDK level dose–response curves previously demonstrated in vitro operate in vivo, but in addition, we show that they are dependent on cell size in vivo (*Pomerening et al., 2003*). We also demonstrate a link between ploidy and CDK activity, with higher ploidy causing a reduction in CDK activity. We propose that CDK activity can be inhibited by a DNA-related mechanism in keeping with early work showing that increasing DNA content delays mitosis, with the removal of DNA by irradiation causing acceleration of the following mitosis (*Devi et al., 1968*; *Sachsenmaier et al., 1970*; *Wilson, 1925*). Our experiments show that DNA inhibits CDK activity more in smaller cells, potentially reducing CDK activity by a titration mechanism. This may be related to the mechanism by which cell size is linked to ploidy across cell types (*Wilson, 1925*; *Amodeo and Skotheim, 2016*; *Prescott, 1956*; *Tzur et al., 2009*). Finally, we show that tyrosine phosphorylation, PP2A activity, and DNA-dependent inhibition of CDK activity act together to restrict small cells from division, forming a mechanism to generate the robust cell size threshold behaviour observed in normal cells.

Our observations suggest that cell size control over the onset of mitosis involves several molecular mechanisms. If it is assumed that the accumulation of the C-CDK cyclin chimera driven by the cyclin promoter mimics the accumulation of cyclin, then one mechanism is the accumulation of cyclin through the cell cycle, which scales with the increase in cell size. A second is a synergistic interaction between the inhibitory CDK tyrosine phosphorylation pathway and the PP2A phosphatase, which acts on both the tyrosine phosphorylation pathway and dephosphorylation of CDK substrates. The third is a DNA-concentration-dependent inhibition of CDK activity. Given the conservation of all

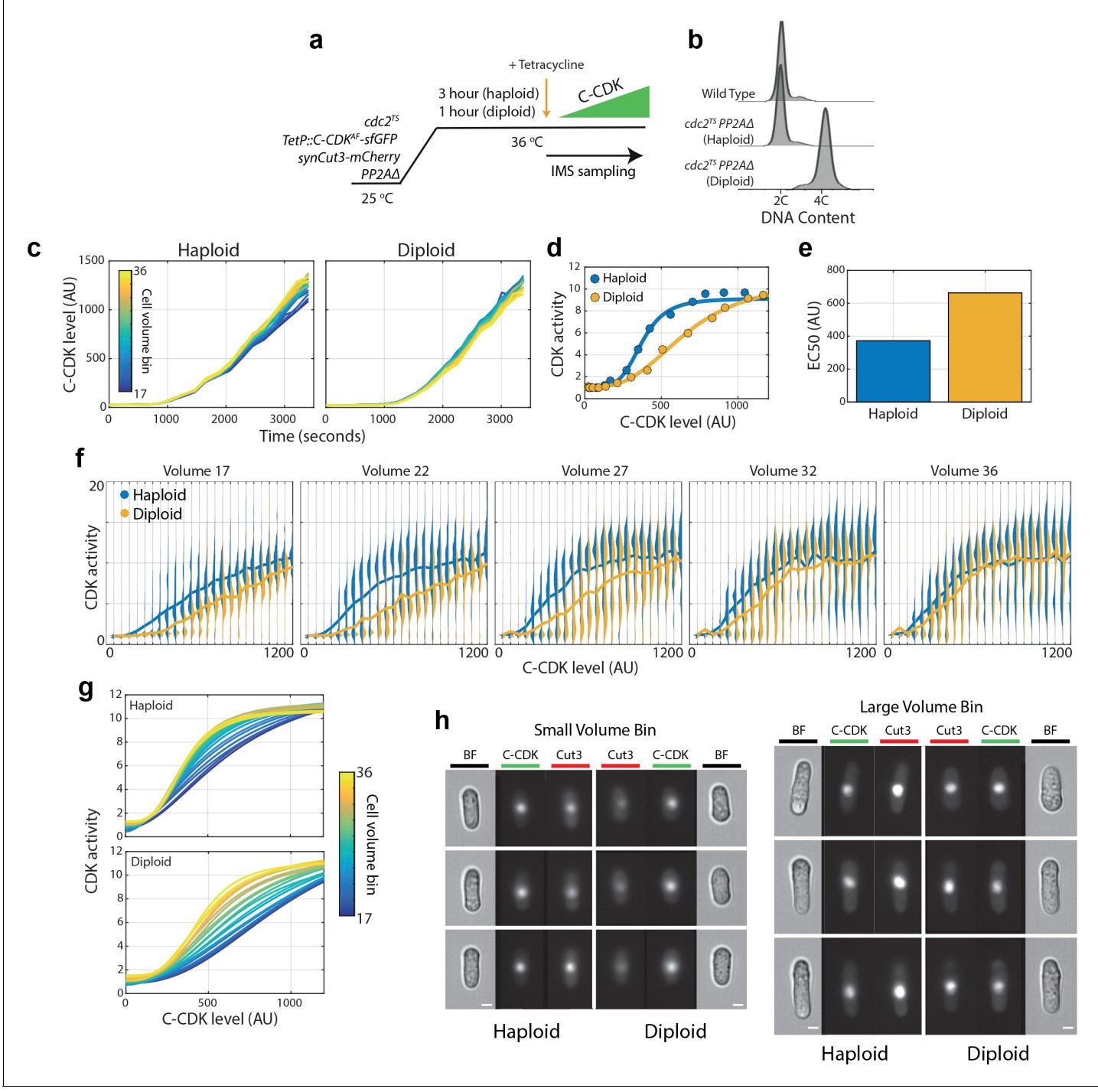

**Figure 4.** Cellular DNA content inhibits CDK activity independently of tyrosine phosphorylation or PP2A activity. (**a**) Experimental outline for panels (**b**)-(**h**). PP2A Δ/Δ diploids and PP2AΔ haploids were arrested using *cdc2^TS^*. *PP2AΔ* refers to a deletion of the *ppa2* gene. Diploids were held at 36°C for 1 hr, whilst haploids were held for 3 hr to generate blocked cell populations with similar cell volumes despite ploidy differences. C-CDK^AF^ expression was induced by addition of tetracycline, and C-CDK^AF^-sfGFP concentration and CDK activity were measured by sequential sampling from time of induction in an imaging flow cytometer. (**b**) Flow cytometric DNA content analysis for wild-type cells, haploid *cdc2-M26 PP2AΔ* cells and diploid *cdc2-M26/cdc2-M26 PP2AΔ/PP2AΔ* cells. *PP2AΔ* refers to a deletion of the *ppa2* gene. Cells were fixed for sampling after the block lengths specified in (**a**), before the addition of tetracycline. (**c**) Expression of C-CDK^AF^ fused to sfGFP from point of tetracycline addition in haploid and diploid strains. Different coloured lines represent different size bins. Haploid n = 125,021, diploid n = 139,557. (**d**) Mean CDK activity against C-CDK^AF^-sfGFP level in haploids and diploids. Solid line is a sigmoid fit to data. (**e**) EC50 from sigmoid curves in (**d**). Haploid EC50: 372 AU. Diploid EC50: 663 AU. Haploid EC50 is 56% of diploid EC50. (**f**) Violin plots of single-cell C-CDK^AF^-sfGFP level against CDK activity in annotated volume bins and ploidy status. Solid line through

*Figure 4 continued on next page*

Figure 4 continued

violin plot indicates the mean CDK activity within the C-CDK-sfGFP level bin. Volume bins span a physiological range of diploid cell sizes. Volume bin 17 corresponds to a haploid cell length of 12.1 µm and a diploid cell length of 9.53 µm. Volume bin 36 corresponds to a haploid length of 18.7 µm and a diploid length of 14.4 µm. (g) Mean intra volume-bin dose–response of C-CDK-sfGFP level vs CDK activity in annotated ploidy level. Lines are sigmoid curves fit to raw data. Cell volume bin indicated by line colour. (h) Example raw images from experiment. Brightfield (BF) channel displaying cell morphology, C-CDK-sfGFP channel and synCut3-mCherry CDK activity indicator are shown. C-CDK level is the same across all images. Scale bars = 3 µm.

these molecular regulators, these mechanisms can be expected to have direct relevance in other eukaryotic cells.

## Materials and methods

### *S. pombe* genetics and cell culture

*S. pombe* media and standard methods are as previously described (*Moreno et al., 1991*). After nitrogen and glucose addition, EMM was filter sterilised. This process allows for the generation of clear un-caramelised media. Nutritional supplements for auxotrophic yeast strains were added at a concentration of 0.15 mg/ml. Temperature-sensitive mutant strains were grown at temperatures as specified in the text. The temperature-sensitive allele of Cdc2 (CDK$^{TS}$) used was Cdc2-M26. To modulate inducible promoters, anhydrotetracycline hydrochloride (Sigma) in DMSO at specified concentrations was added to 0.03125 µg/ml final concentration unless otherwise specified. To alter Cdc2 (as) activity, 1NM-PP1 diluted in DMSO was used at concentrations specified in the text. To stain for septa, calcofluor (Fluorescent Brightener 28 [Sigma-Aldrich]) was made up in water at 1 g/l and stored as 500× stock. SynCut3 was constructed by Gibson assembly of a codon-optimised fragment consisting of the first 528 amino acids of Cut3, a linker region, and a fluorescent protein (mCherry or mNeongreen). YFP was tagged onto C-CDK at the C-terminus of the protein. Where the sfGFP labelled C-CDK was used, the sfGFP was present internally within the Cdc13 component (*Basu et al., 2020*; *Kamenz et al., 2015*). Cut3-mCherry was generated by C-terminal tagging (*Bähler et al., 1998*), and Cut3-GFP was developed previously (*Sutani et al., 1999*). Details of the TetR promoter and linearised variants can be found in a previous publication (*Patterson et al., 2019*). All *S. pombe* strains used in this study are listed in *Table 1*.

### Microscopic imaging

All imaging was performed using a Deltavision Elite (Applied Precision) microscope – an Olympus IX71 wide-field inverted fluorescence microscope with a PLAN APO 60× oil, 1.42 NA objective and a Photometrics CoolSNAP HQ2 camera. To maintain specified temperatures during imaging, an IMSOL imcubator Environment control system and an objective heater was used. SoftWoRx was used to set up experiments. Five z-stacks were acquired, with 1 µm spacing. Image analysis was performed using custom Matlab scripts that executed the steps outlined in *Figure 1—figure supplement 1*.

The ONIX Microfluidics platform allows for long-term time-lapse imaging of live cells. Plate details can be found at http://www.cellasic.com/ONIX_yeast.html. Fifty microlitres of cell culture at density 1.26 × 10$^6$/ml was loaded into the plate and imaged in the 3.5 µm chamber. Cells were loaded at 8 psi for 5 s. Media was perfused at a flow rate of 3 psi. The imaging chamber was washed with media for 1 min at 5 psi before cells were loaded.

Mattek glass bottom dishes were used for some time-lapse imaging applications with drugs that were incompatible with Cellasics plates, primarily for the purpose of release from a 1NM-PP1/Cdc2 (as) cell cycle block. Dishes were pre-treated with soybean lectin to permit cell adherence (Sigma-Aldrich). Before the addition of cells, Mattek dishes were pre-warmed on a heatblock at appropriate temperature. Cells were grown and blocked in liquid culture before 2 ml were pelleted (5000 rpm/ 30 s). Cell pellets were then pooled and resuspended in 1 ml of release media (at which time a stop watch was started) in a new microcentrifuge tube before pelleting (5000 rpm/30 s) and resuspended in 5 µl of media. This concentrated cell suspension was then applied to the centre of the Mattek dish and allowed to settle for ~5 s. The dish was then washed with 1 ml of release media 3×. The dish

**Table 1.** *S. pombe* strains.

| Strain ID | Strain genotype | Source |
|---|---|---|
| JP223 | h? leu1::cdc13P:cdc13-cdc2.as[†]-YFP:cdc13T::ura4 cdc13Δ::natMX6 cdc2::scLeu2 | This work |
| JP224 | h? leu1::cdc13P:cdc13-cdc2AF.as[†]-YFP:cdc13T::ura4 cdc13Δ::natMX6 cdc2::scLeu2 | This work |
| JP670 | h? leu1::cdc13P:cdc13-cdc2.as[†]-YFP:cdc13T::ura4 cdc13Δ::natMX6 cdc2::scLeu2 ura4::Ppcna1-CFP-pcna cut3-mCherry::hphMX6 | This work |
| JP671 | h? leu1::cdc13P:cdc13-cdc2AF.as[†]-YFP:cdc13T::ura4 cdc13Δ::natMX6 cdc2::scLeu2 ura4::Ppcna1-CFP-pcna cut3-mCherry::hphMX6 | This work |
| JP310 | h? leu1::cdc13P:cdc13-cdc2.as[†]:cdc13T::ura4 cdc13Δ::natMX6 cdc2::scLeu2 cut3-tdTomato::hphMX6 | This work |
| JP311 | h? leu1::cdc13P:cdc13-cdc2AF.as[†]:cdc13T::ura4 cdc13Δ::natMX6 cdc2::scLeu2 cut3-tdTomato::hphMX6 | This work |
| JP295 | h? leu1::cdc13P:cdc13-cdc2AF.as[†]:cdc13T::ura4 cdc13Δ::natMX6 cdc2::scLeu2 cut3-GFP::ura4 | This work |
| JP296 | h? leu1::cdc13P:cdc13-cdc2.as[†]:cdc13T::ura4 cdc13Δ::natMX6 cdc2::scLeu2 cut3-GFP::ura4 | This work |
| JP501 | h? cdc2.as[‡]::blastMX6 synCut3-mNeongreen:: leu1+ | This work |
| JP507 | h? cdc2.as[‡]::blastMX6 synCut3-T19V-mNeongreen:: leu1+ | This work |
| JP601 | h? synCut3-mCherry:: leu1 + cut3-GFP::ura4 | This work |
| JP602 | h? cdc2.as[‡]::blastMX6 synCut3-mCherry::leu1 + cut3-GFP::ura4 | This work |
| JP591 | h? cdc2-M26 synCut3-mCherry::leu1 + leu1::enoTetP:cdc13-sfGFP-cdc2.as[†]:adh1T::hphMX6 TetR1[3] | This work |
| JP593 | h? cdc2-M26 synCut3-mCherry::leu1 + leu1::enoTetP:cdc13-sfGFP-cdc2AF.as[†]:adh1T::hphMX6 TetR1* | This work |
| JP603 | h? cdc2-M26 synCut3-mCherry::leu1 + leu1::enoTetP:DBΔcdc13-sfGFP-cdc2.as[†]:adh1T::hphMX6 TetR1* | This work |
| JP605 | h? cdc2-M26 synCut3-mCherry::leu1+ (JPp178) leu1::enoTetP:DBΔcdc13-sfGFP-cdc2AF.as[†]:adh1T::hphMX6 TetR1* | This work |
| JP679 | h? cdc2-M26::blastMX6 synCut3-mCherry::leu1 + leu1::enoTetP:DBΔcdc13-sfGFP-cdc2.as[†]:adh1T::hphMX6 TetR1* ppa2Δ::kanMX6 | This work |
| JP680 | h? cdc2-M26::blastMX6 synCut3-mCherry::leu1 + leu1::enoTetP:DBΔcdc13-sfGFP-cdc2AF.as[†]:adh1T::hphMX6 TetR1* ppa2Δ::kanMX6 | This work |
| SB175 | h? cdc2-M26 synCut3-mCherry::leu1 + leu1::enoTetP: DBΔcdc13-sfGFP-cdc2AF.as[†]:adh1T::hphMX6 TetR1* ppa2Δ::kanMX6 | This work |
| SB176 | h?/h? cdc2-M26/cdc2-M26_ synCut3-mCherry::leu1+/synCut3-mCherry::leu1+ _leu1::enoTetP: DBΔcdc13-sfGFP-cdc2AF.as[†]:adh1T::hphMX6/leu1::enoTetP: DBΔcdc13-sfGFP-cdc2AF.as[†]:adh1T::hphMX6_ TetR1*/ TetR1*_ppa2Δ::kanMX6/ppa2Δ::kanMX6 | This work |

*TetR1 – CMVP:TetOx1:TetR-tup11Δ70 (described originally by **Patterson et al., 2019**).

[†]Cdc2(F84G).

[‡]Cdc2(F84G, K79E).

was then filled with 3 ml of release media before rapid imaging. In general, the wash process required 1.5 min, and imaging setup requires 5 min for ~8 fields of view. The levels of Cut3 and C-CDK referred to in the paper reference the concentration of both these proteins. This was measured by finding the mean intensity of the brightest group of 9 pixels within the cell, to give a measurement of the concentration.

## Imaging flow cytometry

Imaging flow cytometry was performed with an Imagestream Mark X two-camera system (Amnis), using the 60× objective. Cells were concentrated by centrifugation (5000 rpm/30 s) and resuspended in ~25 µl of media before sonication in a sonicating water bath. In focus, single cells were then gated by applying the following gating methods in sequence:

1. Gradient RMS > 65 (a measure of cell focus).
2. Area/aspect ratios consistent with single cells.

To avoid any autofocus based drift within an experiment, cell were imaged at fixed, empirically determined focal points, designed to maximise the number of cells with gradient RMS > 65. Data was analysed using custom Matlab scripts. The steps these scripts executed were similar to the image processing pipeline for wide-field imaging (*Figure 1—figure supplement 1*) albeit slightly simplified given the presence of only a single cell per image. The Imagestream acquires two bright-field images of a cell, allowing definition of a cell region using the standard deviation of pixels between the two. This is analogous to the approach used for time-lapse cell region segmentation (*Figure 1—figure supplement 1*).

A line was then drawn through the middle of the cell, by finding the middle pixel on the horizontal axis of the cell. The line was widened by one pixel either side, and the mean fluorescence intensity within that line extracted. The standard peak finding algorithm within Matlab was then used to identify nuclear Cut3 fluorescence (either a dip in the case of a low CDK activity cell or a peak in the case of a high CDK activity cell). Background or 'cytoplasmic' fluorescence was defined by the intensity of the flat regions of the curve either side of the peak. The levels of Cut3 and C-CDK referred to in the paper reference the concentration of both these proteins. This was measured by finding the mean intensity of the brightest group of 9 pixels within the cell, to give a robust measurement of the concentration.

To perform time-lapse imaging flow cytometry (IMS), water baths at specified temperatures for the experiment were set up with cultures next to the IMS. Time was measured from the point of drug addition to liquid culture or as described during a wash protocol for drug release. Samples were collected as above from the waterbath, and sample timepoints defined as the time at which acquisition on the IMS began (as opposed to time when sample was collected – although this was consistently ~3 min apart). Samples were imaged for ~1 min unless otherwise stated.

## Data analysis and plotting

### Boxplots

The top of box is the 25th percentile of the data, and the bottom is the 75th percentile. The line in the middle of the box is the median. Whisker lengths are either the distance to the furthest point outside of the box, or 1.5× the interquartile range, whichever is lower. If data exists that is greater than 1.5×, the interquartile range from the top or bottom of the box, this is shown as a red '+'.

### Statistical testing

Statistical testing was performed where appropriate using a two-tailed two-sample t-test. p-values below 0.05 were considered significant. Replicates are shown where appropriate by N numbers.

## Cell size measurement

Cell size was measured by three different metrics. In time-lapse microscopy assays, cell size was determined as the area of the 2D surface segmented by our segmentation algorithm. In the high-throughput imagestream assays, cell size was measured as length of the cell. The difference in metric choice between these two systems was due to improved ability of measuring cell length in the high-throughput assay, where it was less affected by focal-dependent changes in cell volume. In the haploid vs diploid experiments, a measure of cell volume was used, where cells were assumed to behave as cylinders, and volume was calculated from the measured radius and length. This was done as diploids are wider than haploids, and thus a simple length metric cannot be employed for size binning.

## Acknowledgements

We thank Jessica Greenwood and Clovis Basier for their extensive efforts in editing the manuscript. This work was supported by the Francis Crick Institute that receives its core funding from Cancer Research UK (FC01121), the UK Medical Research Council (FC01121), and the Wellcome Trust (FC01121). In addition, this work was supported by the Wellcome Trust Grant to PN (grant number 214183 and 093917), The Lord Leonard and Lady Estelle Wolfson Foundation, and Woosnam Foundation. JOP and PR acknowledge the support of the Biotechnology and Biological Sciences Research Council under grant BB/P026818/1. PR also acknowledges the support of the Biotechnology and Biological Sciences Research Council/National Science Foundation under grant BB/N005163/1 and NSF DBI 1458626. JOP acknowledges support from the Boehringer Ingelheim Fonds PhD fellowship. For the purpose of Open Access, the author has applied a CC BY public copyright licence to any Author Accepted Manuscript version arising from this submission.

## Additional information

### Funding

| Funder | Grant reference number | Author |
|---|---|---|
| Boehringer Ingelheim Fonds | | James Oliver Patterson |
| Cancer Research UK | FC01121 | James Oliver Patterson<br>Souradeep Basu<br>Paul Nurse |
| Medical Research Council | FC01121 | James Oliver Patterson<br>Souradeep Basu<br>Paul Nurse |
| Wellcome Trust | FC01121 | James Oliver Patterson<br>Paul Rees<br>Paul Nurse |
| Wellcome Trust | 214183 | James Oliver Patterson<br>Souradeep Basu<br>Paul Nurse |
| Wolfson Foundation | | James Oliver Patterson<br>Souradeep Basu<br>Paul Nurse |
| Biotechnology and Biological Sciences Research Council | BB/P026818/1 | James Oliver Patterson<br>Paul Rees |
| Biotechnology and Biological Sciences Research Council | BB/N005163/1 | Paul Rees |
| National Science Foundation | 1458626 | Paul Rees |
| Wellcome Trust | 093917 | James Oliver Patterson<br>Paul Rees<br>Paul Nurse |

The funders had no role in study design, data collection and interpretation, or the decision to submit the work for publication.

### Author contributions

James Oliver Patterson, Conceptualization, Software, Formal analysis, Validation, Investigation, Visualization, Methodology, Writing - original draft; Souradeep Basu, Investigation, Writing - original draft, Writing - review and editing; Paul Rees, Data curation, Formal analysis, Visualization, Methodology, Writing - review and editing; Paul Nurse, Conceptualization, Supervision, Funding acquisition, Project administration, Writing - review and editing

### Author ORCIDs

James Oliver Patterson https://orcid.org/0000-0003-1993-4500
Souradeep Basu https://orcid.org/0000-0003-4448-8688

**Decision letter and Author response**
Decision letter https://doi.org/10.7554/eLife.64592.sa1
Author response https://doi.org/10.7554/eLife.64592.sa2

## Additional files

### Supplementary files

• Transparent reporting form

### Data availability

Analysed data has been uploaded to Figshare with the handle 10779/crick.14633037.

The following dataset was generated:

| Author(s) | Year | Dataset title | Dataset URL | Database and Identifier |
|---|---|---|---|---|
| Basu S, Patterson JO, Rees P, Nurse P | 2021 | Source data for "CDK control pathways integrate cell size and ploidy information to control cell division" | https://figshare.com/s/bdc29f893e7153c0e859 | Figshare, 10779/crick. 14633037 |

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
