## [Decision Letter]

**Acceptance summary:**

Through technically impressive single-cell experiments and an array of genetic tools, this paper elegantly dissects the pathways that influence CDK1 activity and therefore cell cycle progression. A key question in this field is how cell size influences CDK1 activity, which allows cells to maintain their size. This work shows that CDK1 activity remains influenced by cell size when known pathways are eliminated and makes the novel observation that ploidy influences CDK1 activity.

**Decision letter after peer review:**

Thank you for submitting your article "Synergistic CDK control pathways maintain cell size homeostasis" for consideration by *eLife*. Your article has been reviewed by 3 peer reviewers, one of whom is a member of our Board of Reviewing Editors, and the evaluation has been overseen by Aleksandra Walczak as the Senior Editor. The reviewers have opted to remain anonymous.

The reviewers have discussed the reviews with one another and the Reviewing Editor has drafted this decision to help you prepare a revised submission.

Summary:

In this paper, Patterson et al. use genetic techniques in fission yeast to understand how cell size impacts CDK activity and entry into mitosis. By disrupting several regulators of CDK activation and using single-cell readouts of cell size, CDK level, and CDK activity, they show that several layers of control synergistically contribute to making CDK activation dependent on cell size. Their data suggest that DNA concentration affects CDK activity and prevents entry into mitosis in cells that have not reached a certain size threshold.

The experiments' technical level is impressive. The authors have developed a novel sensor, synCut3-mCherry, as single-cell read-out for CDK1 activity, and, for the first time, experimentally demonstrate bistability of CDK regulation in fission yeast. The results will be of interest to researchers studying the cell cycle and cell size control.

The paper could be strengthened by directly examining size homeostasis (as is claimed in the title). In addition, the reviewers had some concerns that cell cycle state is not taken into account as one more variable that could influence the experimental results, and felt that the authors need to provide additional information both on their methods and on their interpretation of the results to make this paper understandable to a wider group of researchers.

Essential revisions:

(1) The authors do not mention that the cell cycle driven by phosphorylatable and non-phosphorylatable (AF) versions of cyclin-CDK fusion proteins could be very different. The data in Figure 1c/k strongly suggests that this is the case. This is also supported by prior findings (Figure 5 in Coudreuse and Nurse, 2010) despite some differences in the genetic background (CCP deletion). Overall, the data suggest that G1 phase is extended in the AF-carrying strain. For the experiments in Figure 2h, what is the cell cycle distribution (G1/S/G2) in the different size bins for the different strains, and does cell cycle state/DNA content (S vs. G2) influence the results?

At the very least, the differences in cell cycle structure between the strains need to be discussed. In addition, please specify which of the PP2A genes has been deleted, and how this influenced the cell cycle profile.

(2) Figure 4 shows that small diploid cells display a lower CDK activity than haploids of the same size. This suggests that a titration mechanism prevents the increase in CDK activity when DNA concentration is too high. These data would be strengthened by a direct readout of DNA content (e.g. DAPI) to measure the amount/concentration of DNA in single cells and its relation to CDK activity. Since DNA staining will also reveal cell cycle state (S phase vs. G2 phase), this would also allow it to exclude cell cycle effects by binning for G2 cells.

(3) PP2A has a dual effect on CDK activity, which is insufficiently discussed. PP2A regulates inhibitory phosphorylation of CDK through the feedback enzymes (Wee1 and Cdc25) as was shown by Chica et al., 2016 (which should be cited) and PP2A controls the phosphorylation state of CDK substrates. These two effects should be distinguished, because one of them influences the intrinsic CDK activity, while the second has an effect on net CDK phosphorylation of substrates. The authors need to check their text for statements that are not clear enough in that regard, e.g. in line 151 "PP2A and inhibitory tyrosine phosphorylation constitute two fundamentally different modes of lowering CDK activity" is not entirely correct.

(4) The major claim in the title is about cell size homeostasis, but size homeostasis is not examined directly. What is the cell size distribution at division in a C-CDK-AF PP2A-Δ strain and do these cells show size homeostasis (as judged by a "Fantes plot")? In the absence of any such data, the title would need to be revised.

(5) Showing the bistable behavior of CDK activity in Figure 2h is a beautiful experimental result. The authors show that the fusion-protein threshold to activate CDK is cell size dependent. But the authors do not explain why the CDK activity above the threshold is constant rather than increasing with the fusion-protein level on the x-axis. What is the limiting factor for CDK activity above the threshold? Does the sensor become saturated at high activity levels?

Furthermore, these important results would deserve a better presentation: calculating the mean CDK activity for a bimodal distribution seems meaningless and confusing. The main point here is at which threshold of fusion-protein levels the CDK activities bifurcate.

(6) In Figure 1j-o, the Pearson correlation coefficient should be provided to support the conclusions of the authors. On line 101, the authors mention that "C-CDK-AF cells… show size-dependent CDK activity scaling". Yet, in Figure 1m it does look like this dependency is significantly impaired. It is actually unclear why that is, knowing size scales with C-CDK level, and C-CDK levels correlate with CDK activity. Please discuss this.

(7) Either title or abstract should mention the experimental system being used.

(8) The paper is very dense and will be hard to understand for people not very familiar with this field. Please expand the introduction and discussion to make the text more accessible. It would be interesting if the authors could elaborate on the implications of their results regarding the mechanism(s) of size control, how their findings relate to (extensive) existing literature, and potentially other model systems in which these questions have been investigated. The introduction would benefit from citing one of the many excellent, recent reviews on the topic.

(9) More details need to be provided in the Methods section in order to allow other researchers to evaluate, and possibly reproduce, these experiments. For example:

– The authors should specify which of the two pp2a genes has been deleted and what effect it had on the cell cycle.

– The authors should explain how they measured the level (concentration) of the fusion protein which accumulates in the nucleus. It makes a huge difference whether the fluorescence is proportional to the concentration or the number of molecules of the fusion-protein.

– When referring to 'custom Matlab scripts', the authors should at least describe the steps that the scripts are executing.

– Please specify what 'our segmentation algorithm' is.

– Presumably FOV stands for 'field of view' – please spell out.

– Is there text missing between line 247 and 248? It is unclear what points 1 and 2 refer to.

– Which cdc2-as allele was used?

– Please provide the full genotype for "TetR1".

– Are the results based on a single strain for each genotype, or have several strains with identical genotype been analyzed? If so, how similar/different were the results?

– The diploid strains do not seem listed in the strain table. How were they generated?

– Figure 1l/n/m/o: When after release are the measurements taken? Please add information to the legend.

---

## [Author Response]

Essential revisions:(1) The authors do not mention that the cell cycle driven by phosphorylatable and non-phosphorylatable (AF) versions of cyclin-CDK fusion proteins could be very different. The data in Figure 1c/k strongly suggests that this is the case. This is also supported by prior findings (Figure 5 in Coudreuse and Nurse, 2010) despite some differences in the genetic background (CCP deletion). Overall, the data suggest that G1 phase is extended in the AF-carrying strain.

We now state that cell cycles driven by a phosphorylatable and non-phosphorylatable cyclin-CDK fusion proteins differ, with C-CDK-AF cells possessing a longer G1 phase [Line 85-88]. We also add a reference supporting this statement.

For the experiments in Figure 2h, what is the cell cycle distribution (G1/S/G2) in the different size bins for the different strains, and does cell cycle state/DNA content (S vs. G2) influence the results?At the very least, the differences in cell cycle structure between the strains need to be discussed. In addition, please specify which of the PP2A genes has been deleted, and how this influenced the cell cycle profile.

We have conducted additional experiments that ensure cells are uniformly in G2 following the *cdc2^TS^* block. These demonstrate that there are unlikely to be confounding effects due to differential cell cycle state, as all cells are in the same phase. We have incorporated this result into Figure 2 (Figure 2f) and have added an explanation on lines 162-165.

We also now state in the text and figure legends where appropriate that we have deleted the gene encoding the major PP2A catalytic subunit, *ppa2* [lines 161-162, figure legend 2a]. We have also conducted additional experiments to confirm that the deletion of PP2A does not influence the cell cycle profile of fission yeast after the cell cycle block, and present this result as part of the previously mentioned Figure 2f.

(2) Figure 4 shows that small diploid cells display a lower CDK activity than haploids of the same size. This suggests that a titration mechanism prevents the increase in CDK activity when DNA concentration is too high. These data would be strengthened by a direct readout of DNA content (e.g. DAPI) to measure the amount/concentration of DNA in single cells and its relation to CDK activity. Since DNA staining will also reveal cell cycle state (S phase vs. G2 phase), this would also allow it to exclude cell cycle effects by binning for G2 cells.

This is a pertinent suggestion however these experiments are difficult to perform due to technical limitations of the experimental system used. As the reviewers suggest, a stoichiometrically binding DNA dye may be used as a readout for DNA concentration, however the dyes that we have used provide weak signal when in live cells. This problem is compounded when coupled with the ImageStream X imaging flow cytometer we have used for analysis. This machine is able to rapidly sample a large number of cells but, given the rapidity of sampling, lacks the excitation time to sufficiently excite weak fluorophores, thus preventing the use of these DNA binding dyes.

However, we have conducted additional experiments to ensure that the majority of both haploid and diploid cells are in G2 after the cell cycle block. Given that all cells are in G2 at the point of cyclin-CDK induction, we believe that cell cycle effects are largely accounted for. We now include this additional experimental data that demonstrates blocked cells are in G2 as Figure 4b, and provide an explanation on lines 238-240.

(3) PP2A has a dual effect on CDK activity, which is insufficiently discussed. PP2A regulates inhibitory phosphorylation of CDK through the feedback enzymes (Wee1 and Cdc25) as was shown by Chica et al., 2016 (which should be cited) and PP2A controls the phosphorylation state of CDK substrates. These two effects should be distinguished, because one of them influences the intrinsic CDK activity, while the second has an effect on net CDK phosphorylation of substrates. The authors need to check their text for statements that are not clear enough in that regard, e.g. in line 151 "PP2A and inhibitory tyrosine phosphorylation constitute two fundamentally different modes of lowering CDK activity" is not entirely correct.

We designed experiments with phosphorylatable and non-phosphorylatable Cdk1 in backgrounds lacking PP2A to decouple the dual function of the phosphatases mentioned by the reviewer. We did not make this clear in the original text, and apologise for this. We have clarified this issue on lines 45-47.

Furthermore, we have clarified throughout the text that PP2A can exert its influence through both tyrosine phosphorylation and opposing CDK substrate phosphorylation – particularly with regards to the effects of PP2A and its removal [lines 195-198, 262-266, 290-292].

(4) The major claim in the title is about cell size homeostasis, but size homeostasis is not examined directly. What is the cell size distribution at division in a C-CDK-AF PP2A-Δ strain and do these cells show size homeostasis (as judged by a "Fantes plot")? In the absence of any such data, the title would need to be revised.

We agree that a further experiment to illustrate our point would be to detect weaker size homeostasis in a C-CDK^AF^ PP2AΔ strain than either a strain without PP2A or without inhibitory tyrosine phosphorylation. Unfortunately, the C-CDK^AF^ PP2AΔ strain is inviable, and so we were unable to carry out this experiment.

Given that we are unable to directly assess size homeostasis directly, we agree with the reviewer that our title was overstated. Therefore, we have altered the title to “The CDK control network integrates cell size and ploidy status to control cell division”.

(5) Showing the bistable behavior of CDK activity in Figure 2h is a beautiful experimental result. The authors show that the fusion-protein threshold to activate CDK is cell size dependent. But the authors do not explain why the CDK activity above the threshold is constant rather than increasing with the fusion-protein level on the x-axis. What is the limiting factor for CDK activity above the threshold? Does the sensor become saturated at high activity levels?Furthermore, these important results would deserve a better presentation: calculating the mean CDK activity for a bimodal distribution seems meaningless and confusing. The main point here is at which threshold of fusion-protein levels the CDK activities bifurcate.

The reviewers raise a very interesting point that we would like to understand. The CDK activity sensor is probably not saturated above the CDK activity threshold, as the N/C ratio of synCut3 plateau at different levels in different size bins. Although it may be the case that the sensor reaches saturation in the largest size bin, it cannot be the case in smaller size bins. We suspect that some other factor related to size is limiting, such as nuclear/cytoplasmic transport capacity, but we have no evidence to support this or any other explanation for the limiting factor, and would prefer not to speculate about this.

The reviewer is correct that it is inappropriate to calculate a mean of data distributed between two populations. A mean line is appropriate in Figure 2h (now figure 2i) for C-CDK^AF^ expressing strains as these strains show less bistability than C-CDK^WT^. For the C-CDK^WT^ expressing strain we have plotted the average of the two means derived from the two populations because the gradient of this line also provides information about the degree of bifurcation in the C-CDK^WT^ expressing strain. We now explain what we have plotted, and why, for both C-CDK^WT^ and C-CDK^AF^ expressing cells more thoroughly [lines 174-178].

(6) In Figure 1j-o, the Pearson correlation coefficient should be provided to support the conclusions of the authors. On line 101, the authors mention that "C-CDK-AF cells… show size-dependent CDK activity scaling". Yet, in Figure 1m it does look like this dependency is significantly impaired. It is actually unclear why that is, knowing size scales with C-CDK level, and C-CDK levels correlate with CDK activity. Please discuss this.

We now provide Pearson correlation coefficients in the legend to Figure 1j-o.

The reviewers are correct that the size-dependent CDK activity scaling of C-CDK^AF^ is not clear in Figure 1m. This analysis involves 81 cells so we repeated this experiment in a high-throughput assay with >400,000 cells, where size dependent CDK activity scaling of C-CDK^AF^ expressing cells is apparent (Figure 1, figure supplement 5e). We have explained this in the main text, stating that the size dependent activity scaling is less clear in C-CDK^AF^ expressing strains when compared to C-CDK^WT^ (Figure 1m), but is apparent when repeated in a high-throughput assay [lines 129-135].

As for why this relationship is less clear when compared to C-CDK^WT^, we suggest that this is due to a minority of cells at all cell lengths with low amounts of C-CDK^AF^ protein, which impairs the correlation. A population of cells with low C-CDK^AF^ protein level is visible in Figure 1k. However, given this data, we have now deleted the sentence in question, and now only state that C-CDK^AF^ cells can generate a C-CDK threshold for division [lines 136-138]

(7) Either title or abstract should mention the experimental system being used.

We now include the model organism in the abstract [Line 21].

(8) The paper is very dense and will be hard to understand for people not very familiar with this field. Please expand the introduction and discussion to make the text more accessible. It would be interesting if the authors could elaborate on the implications of their results regarding the mechanism(s) of size control, how their findings relate to (extensive) existing literature, and potentially other model systems in which these questions have been investigated. The introduction would benefit from citing one of the many excellent, recent reviews on the topic.

We have now expanded both the introduction [lines 42-74] and discussion [lines 260-269, 275-279, 286-295] extensively, providing more explanation of our experimental rationale and context for our findings. In the introduction, we now elaborate on the previously suggested model of size control in fission yeast [lines 51-58], as well as highlighting experimental results that suggest our current understanding of cell size control is incomplete [lines 55-58]. In the discussion, we place our work in the context of these results [lines 260-269], and suggest that size control is unlikely to arise from any singular molecular control pathway due to cell size feeding into the CDK control network at multiple points [lines 286-295]. In light of the reviewers comments we have included reviews on cell size control in our introduction [references 2,3,4] and more generally improved our referencing of previous work throughout the text. We thank the reviewers for their useful comment that our original text was too dense to be readily understood and required better referencing and discussion. We hope our revised version is sufficiently improved to be acceptable.

(9) More details need to be provided in the Methods section in order to allow other researchers to evaluate, and possibly reproduce, these experiments. For example:– The authors should specify which of the two pp2a genes has been deleted and what effect it had on the cell cycle.

We now specify that we delete the gene encoding the major PP2A catalytic subunit, *ppa2*, in the text [line 161-162], and in the legend to Figure 2,3 and 4. We have also clarified this in the strain table within the methods section.

– The authors should explain how they measured the level (concentration) of the fusion protein which accumulates in the nucleus. It makes a huge difference whether the fluorescence is proportional to the concentration or the number of molecules of the fusion-protein.

The levels referred to in the paper reference the concentration of the fusion protein. This was measured by finding the mean intensity of the brightest group of 9 pixels within the cell, to give a measurement of the concentration. We have added this to the methods section [lines 346-348].

– When referring to 'custom Matlab scripts', the authors should at least describe the steps that the scripts are executing.

For time-lapse microscopy, the steps involved in segmentation are now described in the additional supplementary figure (Figure 1—figure supplement 1) that we have supplied. We have added a reference to this new figure within the methods section [line 325-326] and in the main text [lines 92-93].

For Imagestream image segmentation, the steps were similar albeit slightly simplified given the presence of only a single cell per image. The Imagestream acquires two brightfield images of a cell, allowing definition of a cell region using the standard deviation of pixels between the two. This is analogous to the approach used for timelapse cell region segmentation.

A line was then drawn through the middle of the cell, by finding the middle pixel on the horizontal axis of the cell. The line was widened by one pixel either side, and the mean fluorescence intensity within that line extracted. The standard peak finding algorithm within Matlab was then used to identify nuclear fluorescence (either a dip in the case of a low CDK activity cell or a peak in the case of a high CDK activity cell). Background or “cytoplasmic” fluorescence was defined by the intensity of the flat regions of the curve either side of the peak. We have added this explanation to our methods section [lines 359-375].

– Please specify what 'our segmentation algorithm' is.

We now provide a supplementary figure outlining the exact steps our segmentation algorithm executes (Figure 1—figure supplement 1), and have made reference to this pipeline in the text [lines 325-326].

– Presumably FOV stands for 'field of view' – please spell out.

We have corrected this [lines 345-346].

– Is there text missing between line 247 and 248? It is unclear what points 1 and 2 refer to.

We have corrected this [lines 353-357]

– Which cdc2-as allele was used?

The Cdc2(as) in the context of the C-CDK fusion protein corresponds to the F84G mutation, and F84G, K79E in monomeric Cdc2(as). We now specify this as a footnote in the strain table within the methods section [lines 412-414, with corresponding labels throughout the strain table].

– Please provide the full genotype for "TetR1".

We now specify the full genotype for TetR1 as a footnote to the strain table, and provide a reference for the original publication [line 414].

– Are the results based on a single strain for each genotype, or have several strains with identical genotype been analyzed? If so, how similar/different were the results?

We have conducted these experiments with several different isolates of the same strain, and repeats with the same strain. We obtained similar results with both approaches.

– The diploid strains do not seem listed in the strain table. How were they generated?

We generated diploids by two methods – interrupted mating, and transient MBC treatment of the haploid strain, both of which gave similar results in our experiments. The result presented is from the strain produced by interrupted mating. The absence of the strain from the strain table was a mistake, and is now provided.

– Figure 1l/n/m/o: When after release are the measurements taken? Please add information to the legend.

We have now provided this information in the figure legend to Figure 1g, which provides the experimental schematic for the following experiments.